# Sandbur Drought Tolerance Reflects Phenotypic Plasticity Based on the Accumulation of Sugars, Lipids, and Flavonoid Intermediates and the Scavenging of Reactive Oxygen Species in the Root

**DOI:** 10.3390/ijms222312615

**Published:** 2021-11-23

**Authors:** Zhiyuan Yang, Chao Bai, Peng Wang, Weidong Fu, Le Wang, Zhen Song, Xin Xi, Hanwen Wu, Guoliang Zhang, Jiahe Wu

**Affiliations:** 1Institute of Environment and Sustainable Development in Agriculture, Chinese Academy of Agricultural Sciences, Beijing 100081, China; yangzhiyuan0626@aliyun.com (Z.Y.); baichao37@126.com (C.B.); fuweidong@caas.cn (W.F.); songzhen@caas.cn (Z.S.); 2The State Key Laboratory of Plant Genomics, Institute of Microbiology, Chinese Academy of Sciences, Beijing 100101, China; wangpeng@caas.cn (P.W.); wangle510@foxmail.com (L.W.); 3Beijing Key Laboratory of Captive Wildlife Technologies, Beijing Zoo, Beijing 100044, China; 4The State Key Laboratory of Cotton Biology, Institute of Cotton Research, Chinese Academy of Agricultural Sciences, Anyang 455000, China; 5Beijing Plant Protection Station, Beijing 100029, China; xixin1990@163.com; 6E.H. Graham Centre for Agricultural Innovation (A Collaborative Alliance between Charles Sturt University and the NSW Department of Primary Industries), Wagga Wagga Agricultural Institute, Wagga Wagga, NSW 2650, Australia; hanwen.wu@dpi.nsw.gov.au

**Keywords:** *Cenchrus spinifex* Cav., drought, phenotypic plasticity, flavonoids, reactive oxygen species, antioxidant

## Abstract

The perennial grass *Cenchrus spinifex* (common sandbur) is an invasive species that grows in arid and semi-arid regions due to its remarkable phenotypic plasticity, which confers the ability to withstand drought and other forms of abiotic stress. Exploring the molecular mechanisms of drought tolerance in common sandbur could lead to the development of new strategies for the protection of natural and agricultural environments from this weed. To determine the molecular basis of drought tolerance in *C. spinifex*, we used isobaric tags for relative and absolute quantitation (iTRAQ) to identify proteins differing in abundance between roots growing in normal soil and roots subjected to moderate or severe drought stress. The analysis of these proteins revealed that drought tolerance in *C. spinifex* primarily reflects the modulation of core physiological activities such as protein synthesis, transport and energy utilization as well as the accumulation of flavonoid intermediates and the scavenging of reactive oxygen species. Accordingly, plants subjected to drought stress accumulated sucrose, fatty acids, and ascorbate, shifted their redox potential (as determined by the NADH/NAD ratio), accumulated flavonoid intermediates at the expense of anthocyanins and lignin, and produced less actin, indicating fundamental reorganization of the cytoskeleton. Our results show that *C. spinifex* responds to drought stress by coordinating multiple metabolic pathways along with other adaptations. It is likely that the underlying metabolic plasticity of this species plays a key role in its invasive success, particularly in semi-arid and arid environments.

## 1. Introduction

*Cenchrus spinifex* Cav. (common sandbur) is a perennial grass that has spread worldwide as an invasive weed due to its remarkable adaptability. This allows it to grow in semi-arid and arid environments under conditions of drought, salinity, extreme temperature, and barren soil [1]. There are no natural competitors for this species in China, and it has therefore damaged natural ecosystems and reduced crop yields as well as injuring domestic animals that ingest its burs, which feature up to 40 barbed spines [2]. Physical and chemical control measures, and the deliberate planting of competitors, have achieved only limited success [3,4]. Little is known about the molecular basis of drought tolerance in sandbur, but the identification of proteins that facilitate adaptation to drought could lead to the design of more effective measures for the protection of ecological and agricultural systems in colonized habitats.

Plants can escape, avoid, or tolerate drought stress by adopting different strategies [5,6]. Various response mechanisms have therefore evolved, allowing plants to adapt and survive during long periods of drought [7]. For example, desert plants respond to drought by regulating their phenology (choosing extremely short life cycles) or by osmotic adjustment [8,9]. The latter allows for cell enlargement and plant growth during severe drought stress by keeping the stomata partially open to facilitate CO_2_ assimilation [10].

Proteomics has been widely used to characterize the protein-level response to drought stress in model plants and crops [11,12,13,14,15]. Many drought-responsive proteins have been identified, some of which act directly to improve water use efficiency (e.g., aquaporins), while others protect protein integrity (e.g., late embryogenesis abundant proteins) or act as transcription factors or signaling proteins to regulate the overall drought response such as dehydration-responsive element-binding (DREB) proteins [16]. Plants also respond to drought by regulating their metabolism, accumulating natural products that protect cells from stress or promote water retention. For example, compatible solutes such as betaines, sugars, and amino acids increase the osmotic potential of cells and directly prevent water loss [17]. Furthermore, certain secondary metabolites increase drought tolerance by preventing oxidative damage. For example, plants under drought stress accumulate ascorbate, which modulates the redox state of cells to control guard cell signaling and stomatal movement as well as protecting against reactive oxygen species (ROS) and other free radicals [18,19,20]. Similarly, drought stress can induce the synthesis of flavonoids [21], phenylpropanoids [22,23], and terpenoids [24], all of which act as antioxidants.

A few plant species (including *C. spinifex*) withstand drought stress by inducing a broad range of complex responses that trigger significant morphological changes, a phenomenon known as phenotypic plasticity [4,25,26,27]. Phenotypic plasticity can be regulated at multiple levels including transcriptional reprogramming caused by changes in the availability of transcription factors, epigenetic regulation by DNA methylation, and the post-transcriptional regulation of gene expression by microRNAs. Adaptive metabolism is a common component of phenotypic plasticity, in which plants regulate their metabolic processes in response to stress [28]. However, the phenotypic plasticity responsible for drought tolerance in plants has not been studied in detail [29].

Here, we investigated the molecular mechanisms of drought tolerance in common sandbur by combining proteomics and biochemical analysis, using isobaric tags for relative and absolute quantitation (iTRAQ) together with physiological and biochemical assays. Root samples from three soils differing in moisture content—20% (normal), 10% (moderate drought), and 5% (severe drought)—were used for iTRAQ analysis to identify proteins differing in abundance between normal and drought conditions, and the proteins were functionally annotated using our bespoke bioinformatics pipeline. Our data provide insight into the drought resilience and hence the invasive success of *C. spinifex* and could lead to the development of new strategies to protect natural and agricultural environments from this weedy species. 

## 2. Results

### 2.1. Characterization of C. spinifex Plants under Normal and Severe Drought Conditions

For the initial experiments, we recorded morphological and physiological characteristics in a binary comparison of plants growing under normal conditions (20% soil moisture, the G0 control group) or severe drought conditions (5% soil moisture, the G2 drought treatment group). We found that soil moisture conditions significantly affected the growth and development of *C. spinifex* plants. Under severe drought conditions, the plants developed shorter leaf blades, longer roots, and 2–3 additional tillers compared to plants under normal conditions. The G2 plants were also stunted, and the life cycle was ~2 months shorter compared to the G0 plants (Figure 1A–D). The G2 root/shoot ratio was significantly higher (1.09 ± 0.28) than in the G0 plants (0.24 ± 0.08). Similarly, the G2 carbon to nitrogen (C/N) ratio was 30.65 ± 6.25 compared to the G0 value of 26.09 ± 7.08. Finally, the G2 water use efficiency (WUE) was 8.59 ± 1.51 compared to the G0 value of 3.03 ± 0.96 (Figure 1E,F). 

### 2.2. Proteomic Analysis of C. spinifex Roots under Drought Stress

To understand the molecular basis of the observed differences in phenotype, roots of plants grown under normal conditions (G0, 20% soil moisture) were compared to those subjected to moderate drought (G1, 10% soil moisture) and severe drought (G2, 5% soil moisture) by iTRAQ and liquid chromatography tandem mass spectrometry (LC-MS/MS). From a total of 395,016 spectra, 64,089 spectra were identified, resolving to 15,867 unique peptides and 3428 identified proteins, 2417 of which featured at least two unique peptides accounting for 70.5% of all proteins (Figure 2A, Appendix A). The number of peptides of different lengths followed a log-normal distribution (mostly in the range 5–31) with a mode of 11 peptides and a mean peptide length of 14.55 (Figure 2B). The mean coverage of protein fragments with 95% confidence was 18.09% (Figure 2C). Proteins that differed in abundance between conditions (DAPs) were defined as those with a fold change >1.6 (*p* < 0.05). Accordingly, we identified 385 DAPs in total including 262 in the G1/G0 comparison, 240 in the G2/G0 comparison, and 75 in the G2/G1 comparison (Figure 2D). We found that 135 of these proteins overlapped the G1/G0 and G2/G0 comparisons, 10 of which were also present in the G2/G1 comparison.

Functional annotation by enrichment analysis using the *Panicoideae* database identified a total of 3428 proteins including 3268 GO terms, 2323 COGs, and 2151 proteins in KEGG pathways (Appendix A). 

Heat maps of the DAPs identified by iTRAQ analysis revealed six major groups that showed similar tendencies between G1 and G2, with three upregulated groups (I, II, and III) and three downregulated groups (IV, V, and VI) in each case (Figure 3a). Based on GO terms including biological processes, cellular components, and molecular functions, the total protein content (Figure 3b, green) formed 52 clades whereas the DAPs (Figure 3b, yellow) formed only 46. The six clades without DAPs were mostly associated with the biological process categories metabolic process, cellular process, and response to stimuli, and with the molecular functions antioxidant activity, catalytic activity, structural molecule activity, and transporter activity. Among the cellular functions, the most significant differences were associated with the extracellular region and membrane-enclosed lumen. 

The 385 DAPs (Figure 2D) were assigned to six major clusters based on their functional annotations, which were mostly associated with protein synthesis, transport functions, energy utilization, ROS scavenging, secondary metabolism, and the cytoskeleton, plus additional cluster for other functions and non-annotated proteins (Figure 3c). The DAPs associated with protein synthesis, transport functions, and ROS scavenging were mostly upregulated under drought stress, whereas those related to secondary metabolism were downregulated, which would lead to the accumulation of pathway intermediates (Figure 3c). These data indicate that protein synthesis, transporter activity, primary energy metabolism, ROS scavenging, and the synthesis of intermediate secondary metabolites play key roles in *C. spinifex* drought tolerance. 

### 2.3. High Protein Synthesis and Transporter Activity Increase C. spinifex Drought Tolerance

Our proteomic data showed that 145 DAPs involved in protein synthesis or transport functions were upregulated in response to drought (Figure 4a, Appendix A). The 98 DAPs associated with protein synthesis included ribosome subunits, translation initiation factors and tRNA ligases as well as multiple enzymes required for posttranslational modification (Figure 4a). The 47 DAPs associated with transport included five ABC transporters, eight amino acid transporters, and six inorganic ion transporters (Figure 4a, top). In addition, seven spliceosome proteins were also significantly upregulated in response to drought, indicating that post-transcriptional activity was more elevated under drought stress. These results suggest that enhanced protein synthesis and transporter functions in *C. spinifex* roots protect cells from drought stress. 

More importantly, the expression of 45 ribosomal units, 13 translation initiation factors, seven tRNA ligases, and four elongation factor 1 proteins was upregulated in response to drought stress, possibly leading to the increase in protein synthesis (Figure 4a). We selected a panel of nine genes encoding seven ribosome subunits, a translational initiation factor and an ABC transporter for qPCR analysis to determine the corresponding expression levels (Figure 4b, upper panel; Appendix A). The relative gene expression levels in the G1/G0 and G2/G0 root samples were consistent with the corresponding protein concentrations (Figure 4b, lower panel).

### 2.4. Enhanced Energy Metabolism Increases Drought Stress Tolerance in C. spinifex Roots

Our proteomic data also showed that 72 DAPs associated with energy conversion were upregulated in the roots under drought stress (Figure 5a; Appendix A). These included 21 proteins involved in glucose metabolism, galactose metabolism, and glycolysis as well as 11 proteins involved in fatty acid metabolism such as acyl-CoA synthetase, thiolase, and β-ketoacyl-acyl carrier protein reductase (FabG). The data suggest that the plants reallocate their energy reserves under stress to maintain growth and tolerance [8]. Sugars not only provide energy, but also increase osmotic potential to prevent water loss. We identified 18 DAPs associated with sugar and starch metabolism in KEGG pathways. The total sugar content also increased significantly to 1.5-fold normal levels in G1 plants and to 1.8-fold normal levels in G2 plants (Figure 5b). 

One of the key DAPs associated with fatty acid biosynthesis that was upregulated in response to drought was FabG (Figure 5c). We therefore measured the content of several fatty acids by GC-MS, revealing significant increases in the levels of C16 and C18 fatty acids (Figure 5d). These results confirm that carbohydrate and lipid metabolism is reprogramed under drought stress to increase energy conversion and utilization, and possibly also to modify the structure of cell membranes.

### 2.5. Accumulation of Phenylpropanoid and Flavonoid Intermediates May Protect C. spinifex Roots from Drought Stress

Secondary metabolites such as phenolic acids, flavonoids, lignin, and their intermediates protect plants against various forms of stress [30]. Proteomic analysis revealed 30 DAPs involved in secondary metabolism (Figure 6a). The nature of these proteins indicated that intermediates in the phenylpropanoid and flavonoid pathways are likely to accumulate in G1 and G2 roots, reflecting the upregulation of key enzymes in precursor pathways and the downregulation of enzymes that act downstream (Figure 6b). For example, the upregulation of 4-coumarate-CoA ligase (4CL) [EC:6.2.1.12] would promote the accumulation of phenylpropanoid and flavonoid derivatives such as coumaroyl-CoA, chalcones, coumaroyl quinate, and flavanones (Figure 6b). However, the downregulation of shikimate *O*-hydroxycinnamoyl transferase (HCT) [EC:2.3.1.133], anthocyanin synthase (ANS) [EC:1.14.11.19], caffeic acid 3-*O*-methyltransferase (COMT) [EC:2.1.1.68], and peroxidase [EC:1.11.1.7] would prevent the conversion of this pool of intermediates into anthocyanins and lignin (Figure 6b). We therefore measured the anthocyanin and lignin content of the roots and confirmed that the levels of both metabolites were lower in the G1/G2 roots than the G0 controls (Figure 6c,d). The anthocyanin content of the G0 control roots was 9.3 mg g^−1^ fresh weight (FW), but this dropped to 5.8 and 5.4 mg g^−1^ in the G1 and G2 roots, respectively (Figure 6c). Similarly, the lignin content of the G0 roots declined to 82% in G1 roots and to 78% in G2 roots (Figure 6d). 

### 2.6. The ASC–GSH Cycle Drives ROS Scavenging in C. spinifex in Response to Drought Stress

To avoid oxidative stress, plants have evolved enzymatic and non-enzymatic strategies that contribute to cellular redox homeostasis by modulating intracellular ROS concentrations. Our proteomic data revealed that 36 DAPs associated with ROS scavenging or detoxification are upregulated in response to drought stress (Figure 7a; Appendix A) including six proteins related to ascorbate metabolism, 21 glutathione S-transferases (GSTs), and nine NADH/NADPH-related proteins (Figure 7b; Appendix A). These represent two major antioxidant systems in conjunction with the ascorbate–glutathione (ASC–GSH) cycle and the thioredoxin-dependent network for the protection of cells against ROS [31]. However, some major redox-related enzymes showed no significant change such as ascorbate peroxidase and superoxide dismutase.

To determine whether the ASC–GSH cycle facilitates ROS scavenging in *C. spinifex* roots in response to drought stress, we measured ascorbate and NADH levels in the G0, G1, and G2 roots. Ascorbate was significantly more abundant in the G1 and G2 roots than the G0 control, with increases of 1.7-fold and 2.3-fold, respectively (Figure 7c). The NADH/NAD^+^ ratio in the G1 and G2 roots was also significantly higher compared to the G0 control, with increases of 1.3-fold and 1.8-fold, respectively (Figure 7d). These results suggest that the ASC–GSH cycle may facilitate ROS scavenging to improve drought tolerance in *C. spinifex*. 

### 2.7. The Actin Cytoskeleton Participates in the Response to Drought Stress

Finally, our proteomic data revealed 11 DAPs associated with the cytoskeleton, thus contributing to cell division and cellular structure (Figure 8a). Nine DAPs involved in actin and tubulin synthesis were present at significantly lower levels in the G1 and G2 roots than the controls, but at higher levels in the G2 roots than G1 roots (Figure 8a). The analysis of KEGG pathways showed that actin expression was also downregulated during the formation and internalization of phagosomes in G1 and G2 roots (Figure 8a,b). Enzyme-linked immunosorbent assays (ELISAs) confirmed the depletion of actin under drought stress conditions (Figure 8c). Furthermore, we labeled F-actin with phalloidin, allowing us to determine its distribution in fixed tissue sections by confocal microscopy (Figure 8d). Red fluorescence was apparent in the G0 control, but was much less intense in the G1 and G2 roots, supporting the proteomic data and ELISA results (Figure 8c,d). 

## 3. Discussion

Many plants have evolved physiological, biochemical, and molecular mechanisms allowing them to tolerate drought stress. For example, prickly pear cacti (*Opuntia* Mill) have thick stems, needle-shaped leaves, extensive roots, and large amounts of osmotic sap, allowing them to grow well in arid sand [32]. Other plants can deploy such mechanisms electively, allowing them to adapt to changing environments or to colonize new ones. The resulting diversity of phenotypes arising from the same underlying genome is defined as phenotypic plasticity [33]. *C. spinifex* is a drought-tolerant plant that uses phenotypic plasticity to grow in arid and semi-arid areas, resulting in distinctive morphological and physiological adaptations (Figure 1). However, the biochemical and molecular mechanisms of drought tolerance in this species have not been investigated in detail, and it is well-known that physiological and morphological adaptations often reflect underlying biochemical and metabolic changes [28]. We therefore investigated the drought tolerance of *C. spinifex* by iTRAQ-based proteomic analysis combined with physiological and biochemical assays.

Plants are often subjected to environmental stresses such as drought, salinity, and extreme temperatures, which conspire to reprogram transcription and protein synthesis. Such changes in gene expression and protein accumulation lead to changes in the availability of structural proteins and enzymes, resulting in physiological, cellular, and morphological responses [34]. When *C. spinifex* was subjected to drought stress, we observed the upregulation of many DAPs associated with protein synthesis including ribosomal subunit proteins, indicating that *C. spinifex* accelerates protein synthesis as an adaptive response. Similarly in *Arabidopsis thaliana*, the accumulation of RPS5 protein facilitates tolerance to cold stress [35] and the upregulation of *RPL23aB* and *RPL23aA* transcription increases tolerance to abiotic stress [36]. Ribosomal proteins therefore appear to play a key general role in stress responses by providing the capacity to synthesize other proteins, conferring greater flexibility and adaptability, and this may contribute to the drought tolerance observed in *C. spinifex*. 

We also identified 95 DAPs involved in carbohydrate and fatty acid metabolism, indicating that *C. spinifex* modulates its energy resources in response to drought stress. Similarly, proteomic and metabolic analysis in rice has shown that carbohydrate metabolism plays an important role in cell proliferation and survival under drought stress [14], in part mediated by the accumulation of acetyl-CoA via the tricarboxylic acid cycle to produce more ATP [37]. The results of comparative physiological and proteomic analysis in two poplar species also showed increasing carbohydrate metabolism and energy utilization in response to drought stress [38]. Furthermore, long-term soil moisture stress influences photosynthesis and carbohydrate metabolism in coffee plants [39]. The impact on fatty acid biosynthesis may be direct or may reflect the intrinsic link between carbohydrate and lipid metabolism based on competition for the same precursors [40,41]. For example, starch deficiency in *A. thaliana* resulted in the accumulation of sugars and significantly increased fatty acid synthesis via the modulation of acetyl-coenzyme A carboxylase activity [42], the downregulation of starch synthesis increased fatty acid accumulation in potato and sugarcane [43,44], and starch depletion in the rice *floury shrunken endosperm 1* (*fse1*) mutant increased the lipid content by almost 50% [45]. In *A. thaliana*, adaption to drought stress involves an increase in carbohydrate metabolism to produce soluble sugars that not only supply energy, but also maintain high osmotic pressure and protein stability [46]. Our proteomic analysis showed that *C. spinifex* roots also accumulated more soluble sugar under drought stress than the control conditions, suggesting that the sugars may also be used for the dual purpose of increasing energy availability and protecting cells from osmotic stress (Figure 5). The modulation of fatty acid synthesis can also change the properties of cell membranes and generate bioactive signaling molecules [47].

Plants often synthesize various protective secondary metabolites in response to stress. We found that *C. spinifex* roots under drought stress accumulated more phenylpropanoid and flavonoid intermediates than the control roots, probably reflecting the upregulation of 4CL (which catalyzes the fourth step in the phenylpropanoid pathway) and the simultaneous downregulation of enzymes such as FLS, ANS, HCT, COMT, and POD, which act further downstream. Accordingly, despite the accumulation of intermediates, the anthocyanin and lignin content of *C. spinifex* roots under drought stress was significantly lower than the control roots (Figure 5). Phenylpropanoid and flavonoid intermediates promote homeostasis due to the regulation of enzymes at the branch points of these metabolic pathways in a negative feedback loop [48,49]. These metabolites complement the enzymatic antioxidant system and can inhibit the accumulation of ROS and prevent cell damage [50]. Their biosynthesis and accumulation are generally induced in response to biotic and abiotic stimuli including drought stress [51]. For example, phenolic compounds accumulate in the leaves of the shrub *Cistus clusii* during drought stress [30] and the flavonoid content of motherwort plants also increases in response to drought [52]. In maize leaves, water deficit affects the expression of caffeate *O*-methyltransferase, which is associated with lignin synthesis [53]. However, the secondary metabolites and intermediates that accumulate in plants depend on the type of stress they experience [54]. For example, the amount, type, and localization of flavonols in *A. thaliana* roots and shoots depend on the specific drought conditions [55]. *A. thaliana* plants exposed to drought stress also accumulate antioxidant flavonoids [21]. Yuan et al. [56] reported variation in the accumulation of flavonoid compounds under drought stress in the roots of *Scutellaria baicalensis* Georgi. Our findings suggest that *C. spinifex* roots also accumulate selected phenylpropanoid and flavonoid intermediates that protect the plants from drought stress by the precise modification of metabolic flux. 

Stress responses are often associated with redox homeostasis and normally lead to a rapid and transient increase in the levels of intracellular ROS [57,58]. Plants have evolved an efficient ROS scavenging system to maintain redox homeostasis under stress including enzymatic components such as GSTs [59,60] and metabolic components such as ascorbate and glutathione, forming the ASC–GSH cycle [61,62,63]. This system ensures the efficient removal of hydrogen peroxides and lipid peroxides [64]. Plants exposed to drought stress can accelerate the ASC–GSH cycle to achieve the more efficient detoxification of ROS [58,65], and this is often regulated by abscisic acid, as shown in wheat seedlings exposed to drought [66]. Our proteomic analysis showed that many DAPs associated with the ASC–GSH cycle are upregulated in *C. spinifex* roots including NADH and GST. Accordingly, the NADH and ascorbate levels under drought stress were significantly higher than in control roots (Figure 7). We also carried out experiments to confirm the proteomic data. The results indicated that the ASC–GSH cycle may activate during stress. CAT activity also increased slightly in the *C. spinifex* roots under drought stress (Appendix A). The induction of CAT activity is a common response to drought in many plants [67,68,69]. However, CAT activity declines in some other species including sunflower and wheat [70,71]. Interestingly, we observed no change in various other enzymatic and non-enzymatic antioxidants in our study, in agreement with previous studies showing that the types of antioxidants that accumulate in plants under stress, and the degree of accumulation, is highly dependent on the plant species and the duration or severity of stress [72,73].

Finally, we observed the depletion of F-actin in the *C. spinifex* roots under drought stress consistent with a general downregulation of cytoskeletal proteins (Figure 3c). The reorganization of the dynamic cytoskeleton can protect cells from damage [74,75] and typically involves the formation of stress fibers under the plasma membrane to regulate cell shape and volume [76]. These actin filaments reinforce the cell and are linked to the plasma membrane via actin-binding proteins [77]. The regulation of actins is also used to promote stomatal closure in response to drought stress [78,79] and may be involved in the drought-induced repositioning of plastids [80]. In *A. thaliana* roots, the reorganization of actin filaments has been associated with the production of branched root hairs [81].

## 4. Materials and Methods

### 4.1. Plant Materials and Stress Treatments

The seeds of wild *C. spinifex* Cav. plants were collected from the desert of the Inner Mongolia Autonomous Region of China in September 2015. The seeds were soaked in water at 25 °C for 12 h and sown in commercial pots (diameter = 15 cm, height = 18 cm) containing 2 kg sand, vermiculite, and nutrient soil (2:5:3 volume ratio, after drying the vermiculite and soil at 100 °C) and 400 mL water, making a moisture content of 20%. The pots were placed in a greenhouse (25 °C, 30% relative humidity, 16-h photoperiod, 4000 lux) and the seedlings were irrigated daily to maintain the pot weight at 2.4 kg and the soil moisture at 20%.

For drought stress treatments, 45 *C. spinifex* seedlings (two weeks old) of similar size were assigned to three groups. The control group (G0) was maintained at 20% moisture content by continuing with normal watering as described above. In G1, the soil moisture was reduced to 10% (moderate drought) by maintaining the pot weight at 2.2 kg. In G2, the soil moisture was reduced to 5% (severe drought) by maintaining the pot weight at 2.1 kg. All roots were harvested after 15 days. The drought stress experiments were carried out on three biological replicates, with the samples identified as G0A, G0B, and G0C (replicate controls); G1A, G1B, and G1C (replicate moderate drought); and G2A, G2B, and G2C (replicate severe drought). All root samples were immersed in liquid nitrogen immediately after harvesting and were stored at −80 °C.

### 4.2. Protein Extraction

Total protein was extracted from ~1 g of fresh root tissue by grinding the 15 roots from each sample under liquid nitrogen and sonicating for 15 min with 200 μL tetraethylammonium bromide (TEAB). The extract was centrifuged at 11,000× *g* for 20 min at room temperature and the supernatant was carefully transferred to a fresh tube and precipitated with four volumes of pre-chilled acetone containing 10 mM DTT at −20 °C for 30 min. The acetone was removed by centrifugation as above, and the residue was left at room temperature until the acetone had completely evaporated. The protein mixture was then dissolved in phosphate-buffered saline (PBS, pH 7.4) containing 10 mM DTT and centrifuged as above to remove debris. The supernatant containing the total soluble protein was stored at −80 °C. The total protein concentration was determined using the Bradford assay. 

### 4.3. iTRAQ 

#### 4.3.1. Method for iTRAQ Labeling and Fractionation

The iTRAQ assays were performed at Wuhan Jin Kairui Biological Engineering, Wuhan, China. The protein supernatant prepared by TEAB extraction was thawed, digested with trypsin, and labeled using an 8-plex iTRAQ reagent (AB Sciex UK, Warrington, UK) according to the manufacturer’s instructions. The labeled samples were fractionated by high-performance liquid chromatography (HPLC) using a Thermo DINOEX Ultimate 3000 BioRS system equipped with a Durashell C18 column (5 µm, 100 Å, 4.6 × 250 mm). Twelve fractions were collected.

#### 4.3.2. LC-MS/MS Analysis

The labeled peptides were dried and separated by strong cation exchange chromatography using an AB SCIEX nanoLC-MS/MS (Triple-TOF 5600 plus) system after direct injection onto a 20 μm PicoFrit emitter (New Objective) packed to 12 cm with Magic C18 AQ 3 µm 120 Å resin. The samples were separated in a 2–30% gradient of buffer A (0.1% (*v*/*v*) formic acid, 5% (*v*/*v*) acetonitrile) in buffer B (0.1% (*v*/*v*) formic acid, 95% (*v*/*v*) acetonitrile) over 90 min, and the chromatogram was recorded at 214 nm. MS1 spectra were collected in the range 350–1500 *m*/*z* for 250 ms. The 20 most intense precursors with charge states of 2–5 were selected for fragmentation, and MS^2^ spectra were collected in the range 50–2000 *m*/*z* for 100 ms. Precursor ions were excluded from reselection for 15 s.

#### 4.3.3. Protein Identification and Quantification

The original MS/MS data file was analyzed using ProteinPilot v4.5. For protein identification, we used the Paragon algorithm integrated into ProteinPilot to screen the UniProt *Bos taurus* database (31,872 items, updated in April 2015). The parameters were set as follows: instrument = Triple TOF 5600; iTRAQ quantification; cysteine modified with iodoacetamide; biological modifications = ID focus and trypsin digestion; Quantitate, Bias Correction and Background Correction checked for protein quantification and normalization. An automatic decoy database search strategy was used to estimate the false discovery rate (FDR, ≤0.01) with the Proteomics System Performance Evaluation Pipeline (PSPEP), which is integrated into ProteinPilot. Only proteins with at least one unique peptide and unused values >1.3 were considered for further analysis. Proteins differing in abundance between samples were defined as those showing a fold-change >1.5 and a corrected *p* value (mean value of all compared groups) ≤ 0.05 (*t*-test across all groups). The proteomic data were uploaded to the public repository iProX (ID: IPX0001239001) at URL: https://www.iprox.cn/page/HMV006.html with reference number: IPX0001239001 (accessed on 16 November 2021).

#### 4.3.4. Bioinformatics Pipeline and Annotations

Functional category analysis was performed with Blast2GO v4.5 (http://www.geneontology.org) (accessed on 16 November 2021) with an e-value threshold of 1 × 10^−5^. The best hit for each query sequence was used for Gene Ontology (GO) term matching. We then used the Clusters of Orthologous Groups (COG) database (http://www.ncbi.nlm.nih.gov/COG/) (accessed on 16 November 2021) for the functional annotation of genes from new genomes. Hypergeometric tests were used for the GO and Kyoto Encyclopedia of Genes and Genomes (KEGG) pathway enrichment analysis. 

### 4.4. RNA Extraction and Real-Time PCR

Total RNA extraction and real-time PCR analysis were carried out as previously described for *C. spinifex* [2]. Unique primers were designed for nine candidate genes based on RNA sequencing data, as shown in Appendix A. All data are shown as the mean of three replicate samples ± standard deviations.

### 4.5. Enzyme-Linked Immunosorbent Assay (ELISA)

Proteins associated with drought resistance in *C. spinifex* roots were detected and quantified by ELISA. Protein extracts (10 μg mL^−1^) were transferred to a pre-coated multiwall ELISA plate and incubated at room temperature for 2 h. We prepared blank, negative control, and positive control wells in the same plate. After 6–7 washes with PBS containing 0.1% Tween-20 (PBST), we added 100 µL of the primary F-actin antibodies diluted 1:2000 in PBS (Beijing Genomics Institute, Beijing, China) and incubated at room temperature for 2 h. The plate was then washed 6–7 times with PBST before adding 100 μL of horseradish peroxidase (HRP)-labeled goat anti-rabbit IgG secondary antibodies (diluted 1:5000) and incubated at room temperature for 2 h. After 6–7 further washes with PBST, we added 100 μL fresh 3,3′,5,5′-tetramethylbenzidine (TMB) to each well and incubated the plates in the dark for 20 min. We then measured the OD value at 450 nm using an ELISA plate reader. All data are shown as the mean of three replicate samples ± standard deviations.

### 4.6. Analysis of Total Soluble Sugars

Approximately 1 g of fresh root powder was dried in an oven at 60 °C for two days and the dry weight was determined. We then transferred the powder to 50-mL tubes, added 25 mL of boiled water, and sonicated for 30 min. The resulting suspension was passed through a 0.2 mm filter, the residue was then rinsed three times with distilled water and the liquid filtered as above and pooled with the initial filtrate to make 50 mL in total. We then transferred 1 mL of the filtrate to a fresh tube and added 4 mL 0.2% anthrone before chilling the tube on ice. All samples were then boiled for 10 min, chilled on ice and allowed to warm to room temperature. The sugar content was determined by monitoring the formation of furfural at 620 nm in comparison to standard curves of raffinose according to the following equation: Sugar content = (C × V_2_ × D)/(W × V_1_ × 10)
where C is the sugar content; V_1_ is the determined volume; V_2_ is the extract volume; D is the dilution factor; and W is the sample weight. 

### 4.7. Analysis of the Fatty Acid Content

We mixed 100 mg of fresh root powder with 2 mL 5% (*v*/*v*) hydrochloric acid in methanol to extract polar lipids. We then added 3 mL methanol:chloroform (1:1 *v*/*v*) for the extraction of neutral lipids and 100 μL of the internal standard of 19 methyl acid methyl esters. The mixture was placed in a water bath at 85 °C for 1 h, then cooled to room temperature. We added 1 mL hexane and shook the mixture for 2 min before setting it aside for 1 h to delaminate. We removed 100 μL of the supernatant and mixed it with 900 μL of hexane before passing the mixture through a 0.45-µm membrane. GC-MS was carried out using a Thermo Fisher Scientific (Waltham, MA, USA) Trace1310 ISQ equipped with a TG-5 column (30 m × 0.25 mm × 0.25 μm) to identify 35 lipids. The temperature was set to 80 °C for 1 min, then increased to 200 °C at a rate of 10 °C min^−1^, then increased to 250 °C at a rate of 5 °C min^−1^, and finally increased to 270 °C at a rate of 2 °C min^−1^, with a hold at the final temperature for 3 min.

### 4.8. Analysis of Anthocyanin Levels

Approximately 2 g of fresh root powder was extracted in 100 mL ethanol containing 1% (*v*/*v*) hydrochloric acid as previously described [82]. The absorbance was measured at 535 nm using a UV-2102 spectrophotometer. 

### 4.9. The Klason Method to Determine the Apparent Lignin Content

Approximately 0.5 g of fresh root powder was soaked in 72% sulfuric acid for 3 h, before diluting the sample to 3% sulfuric acid by adding 230 mL water. The mixture was heated for 30 min at 121 °C in an autoclave, cooled overnight, and the Klason residue was recovered by passing the mixture through a glass filter followed by drying at 105 °C in an oven. The quantity of acid-soluble material was determined by UV spectroscopy at 205 nm using a value of 20 for gram absorptivity [83]. The total Klason residue content and acid-soluble materials was expressed as the apparent lignin content.

### 4.10. Analysis of Ascorbate Levels

We mixed 100 mg of fresh root powder with 1 mL 0.2 M hydrochloric acid to extract ascorbate, and determined the quantity as previously described [84]. 

### 4.11. Analysis of Nicotinamide Adenine Dinucleotide (NADH/NAD^+^) Levels

We homogenized 20 mg of fresh root powder in 400 μL NADH/NAD^+^ extraction buffer (Sigma-Aldrich, St Louis, MO, USA) in a microcentrifuge tube and centrifuged the sample at 15,000× *g* for 5 min at room temperature. The supernatant containing NADH/NAD was transferred to fresh tubes in two aliquots of 200 μL, one of which was heated to 60 °C for 30 min to convert NAD^+^ to NADH while the other was set aside. All samples were cooled on ice and centrifuged as above to remove any precipitates. We then transferred 50 μL aliquots of the heated and unheated samples to a 96-well plate in duplicate. We added 100 μL of the master reaction mix to each well (98 μL NAD^+^ cycling buffer, 2 μL NAD^+^ cycling enzyme mix), mixed by shaking or pipetting, and incubated for 5 min at room temperature to convert NAD^+^ to NADH. We then added 10 μL of NADH developer to each well and incubated at room temperature for 1–4 h, depending on color development. The reactions were stopped by mixing the contents of each well with 10 μL stop solution, and we measured the absorbance at 450 nm to determine the NADH/NAD^+^ ratio using the following equation:Ratio = NADH/(NAD_total_ − NADH)

### 4.12. F-Actin Staining and Microscopy 

After 20 days of germination in a growth chamber at 28 °C, the root tips of 3-week-old seedlings were stained with Phalloidin-iFluor 594 (AAT Bioquest, Sunnyvale, CA, USA) to label F-actin. Briefly, a 1000× phalloidin conjugate stock solution was prepared by adding 30 μL of dimethyl sulfoxide (DMSO) to the powder-form vials. A 1× working solution was then prepared by diluting 1 μL of the stock in 1 mL PBS containing 1% bovine serum albumin (BSA). The root tips were stained with the 1× working solution at room temperature for 40 min. The samples were then fixed in 4% formaldehyde in PBS for 10 min. The fixed root tips were incubated in PBS at room temperature for 30 min and rinsed 2–3 times with PBS. The final fixed samples were cut longitudinally into thin sections. Actin filaments were observed under an LSM 700 laser scanning confocal microscope (excitation at 610 nm using a red argon laser) and optical Z-series sections were collected in 0.5 μm steps. 

## 5. Conclusions

The common sandbur *C. spinifex* Cav. shows a remarkable tolerance of drought conditions, which is conferred by its phenotypic plasticity, resulting in distinct morphological changes depending on the availability of soil moisture. Here for the first time, we determined the biochemical and molecular basis of this robust adaptive capability by identifying proteins that accumulate in *C. spinifex* roots specifically under drought stress. The identification of proteins that influence carbohydrate and lipid metabolism as well as major secondary metabolic pathways revealed that *C. spinifex* combines multiple mechanisms to achieve drought tolerance including the accumulation of soluble sugars and lipids, the accumulation of phenylpropanoid and flavonoid intermediates, the acceleration of the ASC–GSH cycle to avoid oxidative stress, and the reorganization of the cytoskeleton. This combination of mechanisms plays a key role not only in the ability of *C. spinifex* to withstand drought, but also its success as an invasive species, allowing it to adapt rapidly to new environments and outcompete native plants. Understanding the molecular and biochemical basis of this phenomenon may help in the development of effective countermeasures against this species to protect agriculture and wildlife.

## Figures and Tables

**Figure 1 ijms-22-12615-f001:**
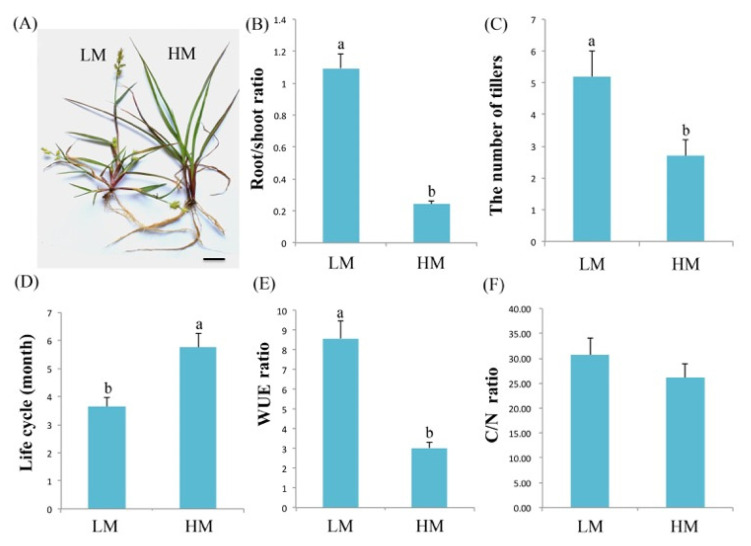
Morphological and physiological characteristics of *C. spinifex* grown under severe drought or normal conditions. (**A**) Morphology of *C. spinifex* plants grown under severe drought (G2, 5% soil moisture) and normal (G0, 20% soil moisture) conditions. (**B**) Comparison of root/shoot ratio. (**C**) Comparison of the number of tillers. (**D**) Comparison of the whole life cycle from germination to seed harvesting. (**E**) Comparison of water use efficiency (WUE). (**F**) Comparison of carbon to nitrogen (C/N) ratio. Different superscript letters indicate statistically significant (*p* < 0.05) differences between G0 and G2 conditions based on the Student’s *t*-test. Differences and standard deviations (error bars) were calculated from the results of three biological replicates. Scale bar = 1 cm.

**Figure 2 ijms-22-12615-f002:**
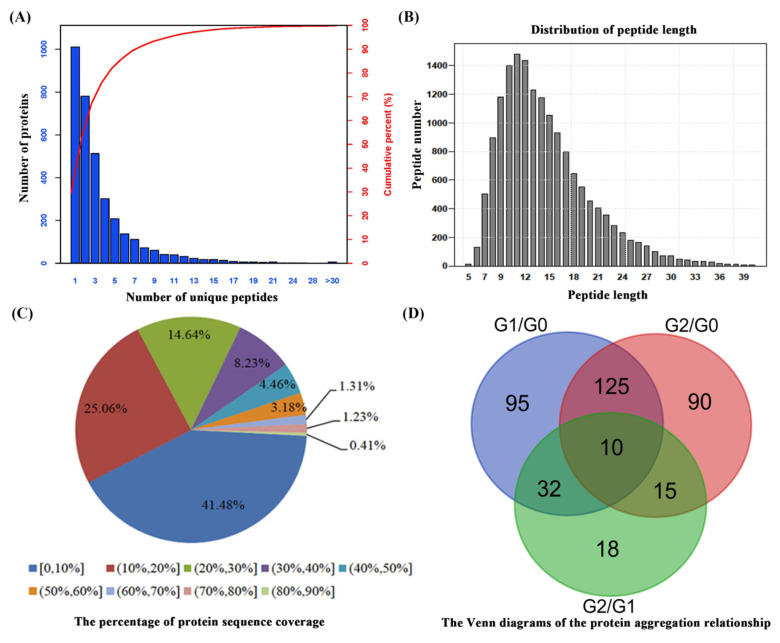
Proteomic analysis of *C. spinifex* roots in response to drought stress. (**A**) The number of unique peptides from the iTRAQ dataset. (**B**) The distribution of peptide lengths. (**C**) The percentage of protein sequence coverage. (**D**) Venn diagrams of the distribution of proteins differing in abundance between the three groups of samples. G0 indicates plants grown under normal conditions (20% soil moisture), G1 indicates plants grown under moderate drought conditions (10% soil moisture), and G2 indicates plants grown under severe drought conditions (5% soil moisture).

**Figure 3 ijms-22-12615-f003:**
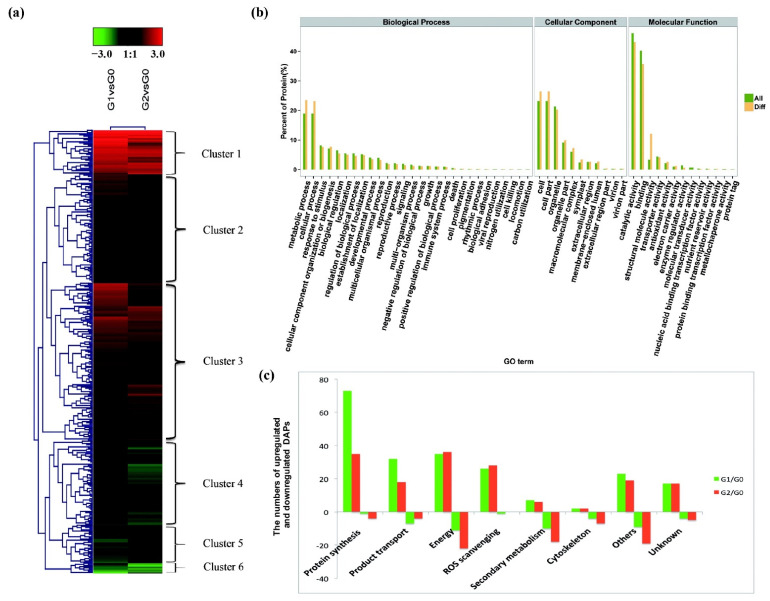
Heat map visualization and clustering of DAP expression profiles in *C. spinifex* roots in response to drought stress. (**a**) The 385 DAPs formed six clusters based on functional annotations (black = no difference to G0 control group, red = upregulation, green = downregulation). (**b**) The functional classification of all the identified proteins (red) and DAPs (yellow) based on Gene Ontology terms. (**c**) The numbers of upregulated and downregulated DAPs in six major functional clusters plus one cluster for other functions and one for non-annotated proteins (green = G1/G0 comparison, red = G2/G0 comparison).

**Figure 4 ijms-22-12615-f004:**
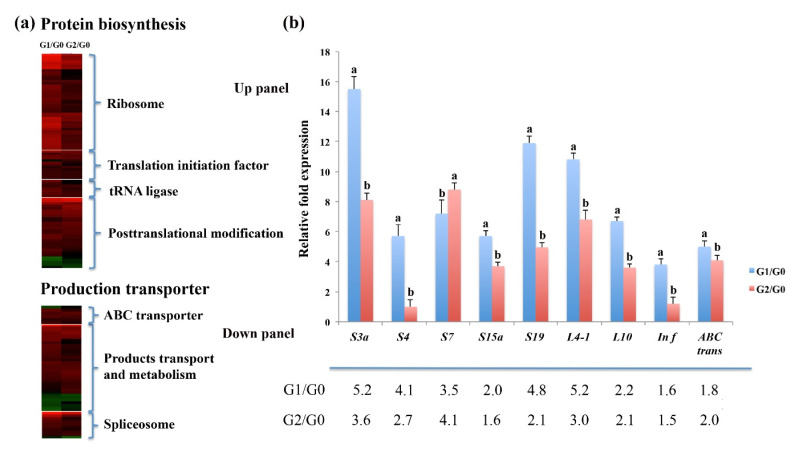
Protein synthesis and transporter functions in *C. spinifex* roots enhance drought tolerance. (**a**) Heat map of protein synthesis and transporter proteins. (**b**) Upper panel: Expression of nine candidate genes encoding DAPs (seven ribosomal proteins, a translational initiation factor and an ABC transporter) analyzed by real-time PCR. Data are means ± standard deviations (*n* = 3). Different superscript letters indicate statistically significant (*p* < 0.05) differences between G1 vs. G0 or G2 vs. G0 based on the Student’s *t*-test. Lower panel: The differential abundance of corresponding DAPs in *C. spinifex* roots in response to drought stress.

**Figure 5 ijms-22-12615-f005:**
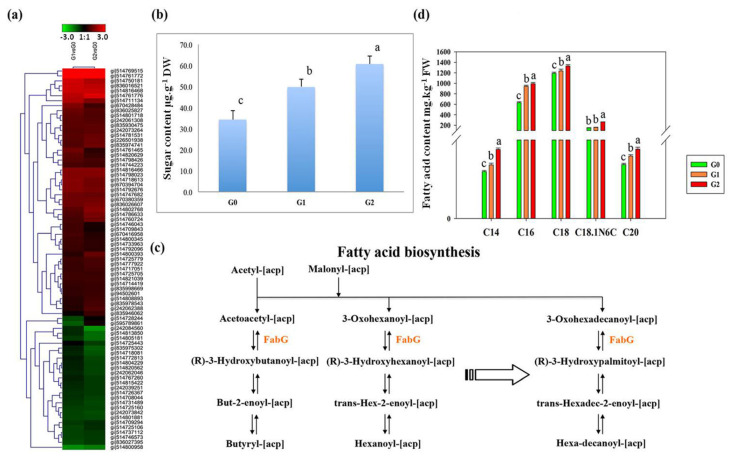
Drought-induced DAPs participate in the biosynthesis of sugars and fatty acids. (**a**) Heat map of DAPs related to energy metabolism. (**b**) Sugar content in the G0 control, G1 and G2 groups. (**c**) Fatty acid biosynthesis pathway. (**d**) Fatty acid content (five different examples) in the control (green), G1 (orange), and G2 (red) groups. The content of each sugar and fatty acid was measured in triplicate, and the data are means ± standard deviations. Different superscript letters indicate significant differences (*p* < 0.05) between the drought treatments and control. DW = dry weight. FW = fresh weight.

**Figure 6 ijms-22-12615-f006:**
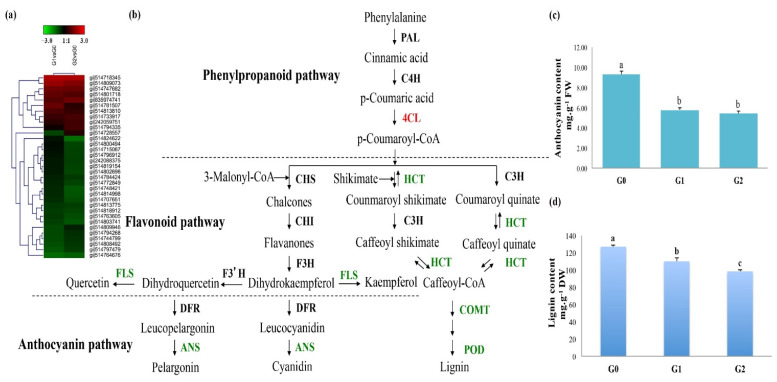
Secondary metabolic pathways involved in *C. spinifex* drought tolerance. (**a**) Heat map visualization of DAPs involved in secondary metabolism. (**b**) Phenylpropanoid, flavonoid, anthocyanin, and lignin pathways, with red indicating enzymes that are upregulated and green indicating those that are downregulated in response to drought stress. (**c**) Anthocyanin content of the control group (G0) and drought treatment groups (G1 and G2). (**d**) Lignin content of the control group (G0) and drought treatment groups (G1 and G2). Anthocyanin and lignin levels are means ± standard deviations of three replicate measurements per group. Different superscript letters indicate a significant difference (*p* < 0.05) between the treatment and control. DW = dry weight. FW = fresh weight.

**Figure 7 ijms-22-12615-f007:**
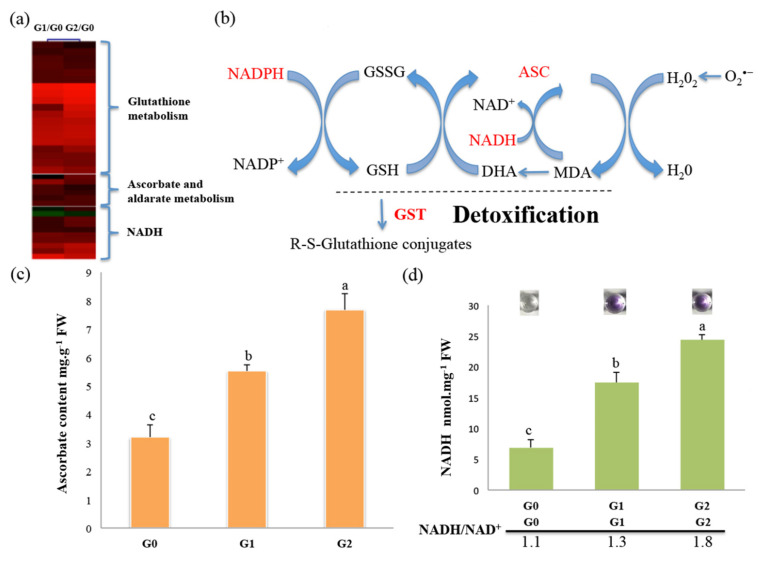
Protein landscape of the ASC–GSH cycle in *C. spinifex*. (**a**) Heat map visualization of DAPs involved in ROS scavenging in response to drought stress. (**b**) The ASC-GSH cycle. (**c**) Ascorbate content of control roots (G0) and those subjected to drought treatment (G1 and G2). (**d**) Upper panel: NADH quantification. Lower panel: The ratio of NADH and NAD^+^. The small images show colorimetric reactions for NADH quantification corresponding to the bars below. Data are means ± standard deviations (*n* = 3). Different superscript letters indicate significant differences between the treatments and the control (*p* < 0.05). Abbreviations: ASC = ascorbate, DHA = dehydroascorbate, MDA = monodehydroascorbate, GR = glutathione reductase, GSH = glutathione, GSSG = oxidized glutathione, GST = glutathione S-transferase. FW = fresh weight.

**Figure 8 ijms-22-12615-f008:**
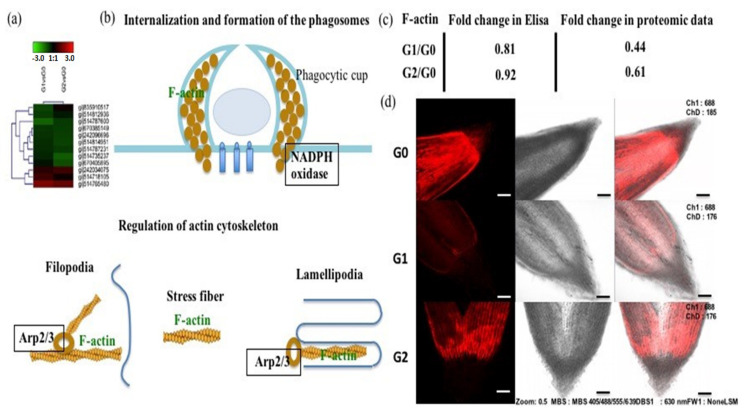
Distribution of F-actin in *C. spinifex* root tip cells under drought stress conditions compared to unstressed controls. (**a**) Heat map visualization of DAPs involved in the cytoskeleton. (**b**) F-actin regulates the internalization and formation of phagosomes and the actin cytoskeleton. (**c**) Abundance of F-actin based on ELISA results and proteomic data. (**d**) Root tip cells stained with phalloidin, showing the distribution of F-actin (scale bar = 10 μm).

## Data Availability

The data that support the findings of this study are available in iProX at URL: https://www.iprox.cn/page/HMV006.html, Reference number: IPX0001239001 (accessed on 16 November 2021).

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
