# Peer review of "Sandbur Drought Tolerance Reflects Phenotypic Plasticity Based on the Accumulation of Sugars, Lipids, and Flavonoid Intermediates and the Scavenging of Reactive Oxygen Species in the Root"

_ijms, 2021, doi:10.3390/ijms222312615_

Round 1
Reviewer 1 Report
This is a complex study consisting of multiple approaches in exploring the drought tolerance of roots of invasive species Cenchrus spinifex. Metabolic pathways routes are complemented with antioxidant defense mechanism, energy transfer, gene expression, and reorganization of cytoskeleton under drought. The study is especially interesting because it is dealing only with the root system under water deficiency and the findings on root „behavior “ under drought are less explored than upper parts of the plant. In my opinion, the title should contain the term „root“.
- The Abstract is well written and reflects objectives and results obtained.
- The Introduction part is well written.
Therefore, I recommend the article for publication.

Author Response
Author response: We are grateful for this comment and have amended the title as follows: “Sandbur drought tolerance reflects phenotypic plasticity based on the accumulation of sugars, lipids and flavonoid intermediates and the scavenging of reactive oxygen species in the root”
Reviewer 2 Report
This papers contains some physiological and molecular aspects of drought tolerance. However, I am wondering why this plant species was considered? Is there any economic value or widely available globally?
Accumulation of sugars, lipids and flavonoid intermediates and the scavenging of ROS improved plant tolerance to drought is well known. What is the novelty in this work and which new parameters are studied in this research is not feasible.
The authors only used one level of drought by which a precise response can not be observed. Drought is a complex stress and it greatly alter in course of time and level of exposure. Therefore, several drought treatment is suggested for such a study.
Although they have measure ROS metabolism but many important enzymes and ROS indicators are not recorded in this study. The study must be complete.
Figures are not professional and self-explanatory. Y-axis should contain full unit of the parameters, not the short form.
Some data could be placed as supplementary file,
Overall, this paper is too wordy. The authors must make it concise.
Although there are many references but vital references on ROS metabolism and antioxidant defense are not read or discussed. Please read 2020-2021 papers on such aspect. Some of the references are old.
Author Response
This papers contains some physiological and molecular aspects of drought tolerance. However, I am wondering why this plant species was considered? Is there any economic value or widely available globally?
Author response 1: Common sandbur is not an economically valuable species in the same way that a crop would be considered economically valuable, but it could be argued that, as an invasive species, it threatens the economic value of crops by behaving as a weed. However, the main reason we considered this species is that it is renowned for its remarkable phenotypic plasticity, which it allows it to withstand severe drought and therefore grows sturdily in arid regions. This is worthy of investigation for two main reasons: (1) defining the mechanism could reveal weaknesses that could be exploited for weed control; and (2) defining the mechanism could reveal strengths that could, in the future, be transferred to crops to increase their resilience. This is already briefly explained in the abstract and introductio
Accumulation of sugars, lipids and flavonoid intermediates and the scavenging of ROS improved plant tolerance to drought is well known. What is the novelty in this work and which new parameters are studied in this research is not feasible.
Author response 2: The reviewer is correct that all these mechanisms have been reported before. However, it is interesting and arguably novel to see all these mechanisms combined and used so effectively in one species, thus underpinning its unusual natural tolerance of arid environments. It is clear notion that there has to be room in the literature for studies that evaluate and report on known mechanisms in species that have not been studied before.
The authors only used one level of drought by which a precise response can not be observed. Drought is a complex stress and it greatly alter in course of time and level of exposure. Therefore, several drought treatment is suggested for such a study.
Author response 3: Sorry, we are puzzled by the comment. In fact, we applied three different treatments: normal (20% moisture) vs two levels of drought (10% moisture = moderate and 5% moisture = severe). Two treatments were only used for the physiological experiments (Figure 1), whereas three treatments were used for the proteomics analysis, which is the bulk of the work reported. We took samples from all three sets of corresponding plants for analysis in order to identify the differentially expressed proteins. This is a rigorous experimental design that has been used in many other studies, so additional experimental groups or time points are not required.
Although they have measure ROS metabolism but many important enzymes and ROS indicators are not recorded in this study. The study must be complete.
Author response 4: Sorry, we do not understand the reviewer comment. In Results section, we have interpreted that there are 36 DAPs associated with ROS scavenging or detoxification including six ascorbate metabolism protein, 21 GSTs and nine NADH/NADPH-related proteins (Fig. 7a and 7b; Table S1). However, some major redox-related enzymes showed no significant change, such as ascorbate peroxidase and superoxide dismutase. We stated it in discussion section as “Interestingly, we observed no change in various other enzymatic and non-enzymatic antioxidants in our study, in agreement with previous studies showing that the types of antioxidants that accumulate in plants under stress, and the degree of accumulation, is highly dependent on the plant species and the duration or severity of stress”. Moreover, we measured ascorbate and NADH levels in the three treatment roots to determine ASC-GSH cycle and NADH function in ROS scavenging. The purpose of the experiments was to determine whether ROS metabolism is one of the mechanisms of drought tolerance in sandbur, and with that in mind our experiments were successful. Our aim was not a comprehensive analysis of antioxidants and the corresponding enzyme cycles.
Figures are not professional and self-explanatory. Y-axis should contain full unit of the parameters, not the short form.
Author response 5: It is acceptable to use abbreviations on graphs if these abbreviations are accepted by convention (e.g., SI units and their derivatives do not have to be explained) or if they are defined in the legend. For example, in Figure 1 we have explained WUE in the legend, so no further explanation is required on the graph. However, we agree with the reviewer that some of the panels could be labeled more informatively, and have checked carefully through the figures to ensure that the axis labels are fully explained as necessary within the bounds mentioned above. We have therefore added an explanation of the carbon to nitrogen (C/N) ratio to Figure 1, and DW = dry weight and/or FW = fresh weight to the legends of Figures 1, 5, 6 and 7. We have added an axis label to Figure 3c. And in Figures 5, 6 and 7 we have added more information to show what is being measured in each graph.
Some data could be placed as supplementary file, Overall, this paper is too wordy. The authors must make it concise.
Author response 6: With respect, this is a subjective opinion not an objective fact. The paper has been revised by a professional scientific writer based in the UK and the amount of text is commensurate with the amount of data presented. Furthermore, the quality of presentation was praised by the other reviewer. We do not see the need for further revision.
Although there are many references but vital references on ROS metabolism and antioxidant defense are not read or discussed. Please read 2020-2021 papers on such aspect. Some of the references are old.
Author response 7: We are grateful for this comment, and we have looked into the literature and provided some more recent references as suggested. Please see discussion section on page 16.
Round 2
Reviewer 2 Report
The revised version is improved.